# Gelatin-Alginate Coacervates Optimized by DOE to Improve Delivery of bFGF for Wound Healing

**DOI:** 10.3390/pharmaceutics13122112

**Published:** 2021-12-07

**Authors:** ByungWook Kim, Eunmi Ban, Aeri Kim

**Affiliations:** Department of Pharmacy, College of Pharmacy, CHA University, Seongnam-si 463-400, Korea; byungwook.kim@genexine.com (B.K.); emban@korea.com (E.B.)

**Keywords:** diabetic foot ulcer, gelatin A, alginate, coacervates, growth factor, bFGF, DOE

## Abstract

Metabolic disorders in diabetic patients are associated with altered protein and lipid metabolism and defects in granulation tissue formation, resulting in non-healing wounds such as diabetic foot ulcers (DFU). Growth factors have essential roles in tissue re-epithelization and angiogenesis during wound healing. In this study, a complex coacervate was evaluated as an enhanced delivery system for fibroblast growth factor (bFGF) to control its release rate and protect it from proteases. Coacervates composed of gelatin Type A (GA) and sodium alginate (SA) were optimized by the Design of Experiments (DOE), with the polymer ratio and the medium’s pH as the independent variables, and turbidity, particle size, polydispersity index, and encapsulation efficiency (EE, %) as the responses. The optimized coacervate protected bFGF from trypsin digestion and showed controlled release compared with bFGF in solution or a physical mixture of GA and SA. It enhanced the viability, migration, and procollagen I C-terminal propeptide synthesis of human dermal fibroblasts in hyperglycemic conditions. In summary, the DOE approach was successfully applied to optimize bFGF GA-SA coacervates as a potential novel therapeutic modality to treat DFU.

## 1. Introduction

Diabetic foot ulcers (DFU) are one of the major complications of diabetes mellitus (DM), with a global prevalence of 3–13% in diabetic patients [1,2,3]. The International Diabetes Federation estimates that at least one limb is lost due to DFU somewhere in the world every 30 s [3]. DFU (non-healing wounds with recurrent injuries) is a complex problem caused by reduced blood supply, peripheral neuropathy, or impaired resistance to infection [4]. According to the DFU clinical guidelines, analgesics and antimicrobial agents are the main pharmacotherapy, with additional wound management such as wound dressing [5]. These treatments have critical limitations: they can only delay the progress of disease and cannot induce tissue regeneration such as re-epithelization and granulation tissue formation. Many in vitro and in vivo studies have been performed to investigate growth factors as the therapeutic agents in impaired wound healing [6]. There are topical solutions or gel products for DFU in the current market: FIBLAST, Easyef, and Regranex containing FGF, EGF, and PDGF, respectively [6]. However, when the growth factors are delivered in solution or as gel formulations, they are prone to degradation at the application site because of the elevated proteases in chronic wounds such as DFU [7].

In our previous studies, we evaluated complex coacervates composed of gelatin Type A (GA) and sodium alginate (SA) as a delivery system to protect the cargo protein from proteases. Complex coacervates are colloidal droplets induced by liquid–liquid phase separation in a solution containing polycation and polyanion [8]. BSA encapsulated in the coacervates of high-molecular-weight Type A gelatin (HWGA) and sodium alginate (SA) was effectively protected from trypsin digestion [9]. However, encapsulation of EGF was not successful in HWGA-SA coacervates. Instead, EGF encapsulation was possible with the coacervates of low-molecular-weight Type A gelatin (LWGA) and SA [10]. EGF in LWGA-SA coacervates was protected from trypsin digestion, showing enhanced activity in diabetic wound healing compared with EGF in a solution or a physical mixture of LWGA and SA [11]. Major factors governing the coacervation process include the temperature, the pH of the reaction medium, the polymer ratio, the molecular weight, the total concentration, and charge densities [12]. One significant difference between HWGA and LWGA was their phase diagrams when mixed with SA in the aqueous phase: the narrow coacervate phase of HWGA and SA mixtures was attributed to strong H-bond interactions [10]. Together with our work on the encapsulation of BSA and EGF, we concluded that the composition of GA and SA coacervates needs to be customized depending on the specific cargo protein.

In this study, we aimed to develop coacervates for delivery of the basic FGF (bFGF), which plays a vital role in wound healing by increasing fibroblast proliferation and migration. Delivery systems for bFGF such as poly(d,l-lactic-co-glycolic acid) nanoparticles and chitosan-fucoidan nanoparticles have been reported for arteriogenesis and nerve tissue engineering, respectively [13,14]. Both systems showed a sustained release of bFGF and protected it from degradation by enzymes. We hypothesized that bFGF encapsulation in the GA-SA coacervates would be affected by differences in the phase behavior between LWGA and HWGA in the presence of SA. We have applied the Design of Experiments (DOE) methodology to optimize HWGA-SA complex coacervates. The pharmaceutical Quality by Design (QbD), now an integral part of pharmaceutical research and development, is defined in the ICH Q8 guidelines as a systematic approach that begins with predefined objectives and emphasizes product and process understanding and process control [15]. DOE is a useful statistical tool for analyzing various elements in applying the QbD principles [16], resulting in less batch failure and more effective and efficient control of the composition or process changes. Protection of the encapsulated bFGF in the optimized HWGA-SA coacervate was tested by a trypsin digestion assay and in vitro release tests. Their effects on the viability and migration of human dermal fibroblasts (HDF) in hyperglycemic conditions were evaluated as an in vitro model to mimic diabetic wounds.

## 2. Materials and Methods

### 2.1. Materials

Gelatin A with gel strength 90–110 (low-molecular-weight gelatin (LWGA), 20–25 kDa on average), Gelatin A with gel strength 300 (high-molecular-weight gelatin (HWGA), 50–100 kDa on average), and citric acid were purchased from Sigma–Aldrich (St Louis, MO, USA). Sodium alginate (SA, JUNSEI, Tokyo, Japan) and acetic acid were purchased from Dae Jung (Siheung-si, Gyeonggi-do, Korea); 0.25% trypsin-EDTA and fetal bovine serum (FBS) were purchased from Gibco (Paisley, UK). Other reagents and their suppliers were as follows: fluorescein-5-isothiocyanate (FITC) from TCI (Tokyo, Japan), silver staining kit from Thermo Scientific, basic FGF (bFGF) from CHA Meditech (Seongnam-si, Gyeonggi-do, Korea), Dulbecco’s phosphate-buffered saline (DPBS) from Welgene (Gyeongsan-si, Gyeongsangbuk-do, Korea), Dulbecco’s Modified Eagle Medium (DMEM) from Hyclone (Logan, UT, USA), the Cell Counting kit-8 (CCK-8) from Dojindo (Kumamoto, Japan), Procollagen I C-terminal propeptide (PICP) ELISA kit from Aviva systems biology (San Diego, CA, USA), HDF cells from Lonza (Walkersville, MD, USA), and the primary antibody of anti-CD31 (ab28364) and rabbit anti-mouse horseradish peroxidase (HRP) antibody as the secondary antibody from Abcam (Cambridge, UK).

### 2.2. Characterization of HWGA and LWGA

#### 2.2.1. Circular Dichroism (CD)

LWGA and HWGA solutions were prepared at a concentration of 5 × 10^−2^ g/L in distilled water (DW) and stored at room temperature (RT) for 24 h. Samples were analyzed by a circular dichroism unit (J-815, Jasco, UK) at RT.

#### 2.2.2. Zeta Potential

A dynamic light scattering instrument (DLS, Zetasizer Nano S, Malvern, UK) was used to measure the pH dependence of the zeta potential. The pH of 1% HWGA and 1% LWGA solutions was adjusted by titration with 0.1 N HCl or 0.1 N NaOH solutions.

### 2.3. Fabrication and Characterization of LWGA-SA and HWGA-SA Coacervates

#### 2.3.1. Fabrication of Complex Coacervates

One percent LWGA (*w/w*) or 1% HWGA (*w/w*) and 1% SA (*w/w*) in DW solutions were used to prepare GA-SA coacervates at various ratios of GA to SA. Varying volumes of 0.5 M acetic acid were added to the GA solutions, and then the SA solution was added to induce coacervation. For encapsulation of bFGF, a bFGF solution was added into acidified GA solutions before adding the SA solution. Turbidity at 450 nm was measured with a UV spectrophotometer (UV-1800, Shimadzu, Japan). Particle size and the polydispersity index (PDI) were analyzed by DLS. Details of the experiment were described previously [9]. The concentration of bFGF or FITC-labeled bFGF (FITC-bFGF) varied depending on the purpose of the experiment, as described in each method. The drug loading efficiency represents the ratio of bFGF to the polymers by weight.

#### 2.3.2. Freeze-Drying of Complex Coacervates

Depending on the purpose of the experiments, 100 μL samples in 96 well-plates or 1 mL samples in centrifuge tubes were freeze-dried in a freeze-dryer (alpha 1-2 LD plus, Martin Christ, Germany) after freezing overnight at −80 °C in a deep freeze. The primary drying was performed at −30 °C and 0.37 mbar for 15 h, and the second drying was performed at −80 °C and 0.001 mbar for 2 h. Images of a cross-section of freeze-dried coacervates were taken by a scanning electron microscope (SNE-4500 M). Adhesive carbon tape was attached to stainless steel stubs. The samples were cut vertically to expose the internal surfaces and placed on the stub and coated with platinum by using Magnetron coating M/C (MCM-100). The samples were then placed in the chamber of the SEM. The freeze-dried samples were kept at −20 °C in heat-sealed aluminum pouches until use.

#### 2.3.3. Trypsin Digestion Assay of Freeze-Dried bFGF Complex Coacervates

A trypsin digestion assay was performed as described previously [9,10,11]. The sample composition was the same for all test samples, with a total polymer weight of 4 mg, a GA to SA ratio of 1:1, and a bFGF content of 10 μg. After freeze-drying, the samples were rehydrated in DW (100 μL), and the coacervates were concentrated by centrifugation (14,000 rpm, 10 min, 4 °C), then 0.25% trypsin-EDTA in DPBS (200 μL) was added to the sample. The samples were incubated in a shaking incubator at 37 °C and 150 rpm for 2 h or 4 h. SDS-PAGE was carried out with a silver staining kit for qualitative analysis of bFGF after the incubation.

#### 2.3.4. Encapsulation Efficiency of Complex Coacervates

FITC and bFGF at a ratio of 4:1 were conjugated according to the Sigma–Aldrich protocol. FITC-bFGF coacervate samples were made at a 1:1 ratio of HWGA/SA (4 mg) with 4 ng of bFGF. Complex coacervates encapsulating FITC-bFGF were separated by centrifugation (14,000 rpm, 10 min, 4 °C), and the supernatant was separated once more by the same centrifugation protocol. The fluorescent intensity in the supernatant was measured at 490 nm for excitation and 525 nm for emission in the SynergyMx microplate reader (Biotek Instruments, Winooski, VT, USA). The percentage of encapsulation was calculated according to the equation below (Equation (1)), assuming that the fluorescence in the supernatant was due to the free FITC-FGF:% Encapsulation efficiency = [(Total FITC-bFGF − FITC-EGF in supernatant)/Total FITC-FGF] × 100(1)

#### 2.3.5. In Vitro Release Test of Freeze-Dried bFGF Complex Coacervates

In vitro release tests of various coacervates were conducted using the Transwell insert method as described previously [9,17]. Freeze-dried HWGA-SA and LWGA-SA coacervates samples with a total polymer weight of 4 mg and a GA to SA ratio of 1:1, encapsulating the 4 ng of FITC-bFGF, were put into the Transwell inserts, and 100 μL DW was added for rehydration of the samples. A release medium (900 μL DPBS) was placed in the wells of 24-well plate as the receiver phase. As a control sample, a FITC-bFGF solution was placed in the Transwell insert. The samples were incubated in a shaking incubator at 37 °C and 150 rpm. Transwell inserts containing samples were moved to the next well containing fresh DPBS at 2, 4, 6, 8, 12, and 24 h to collect the released FITC-bFGF at each time point. The fluorescent intensity in each well was measured at 490 nm for excitation and 525 nm for emission in the SynergyMx microplate reader (Biotek Instruments, Santa Clara, CA, USA).

#### 2.3.6. Phase Diagram of HWGA-SA

Solutions of HWGA (1%) and SA (1%) were mixed at three different ratios (1:0.5, 1:1, 0.5:1) to construct phase diagrams. Varying volumes of 0.5 mM acetic acid were added to each mixture: the acetic acid concentration in the final mixture was in the range of 0~20 mM. Sample turbidity, particle size, and polydispersity index (PDI) were measured as described in Section 2.3.1.

### 2.4. Design of Experiment (DOE)

Coacervates composed of HWGA, SA, and acetic acid (AA) were selected for further optimization based on better protection of bFGF from proteolysis in HWGA coacervates compared with LWGA coacervates. To evaluate the effects of the three variables on the characteristics of the coacervate, and to optimize the coacervate composition, Box–Behnken response surface model analysis was carried out using Minitab 16 software (Version 16.2.3, Minitab LLC, Coventry, UK). The low and high levels of each independent variable were based on the preliminary test results. In total, 15 experimental runs were constructed for the three variables at three levels (−1, 0, +1) of each variable.

Fifteen samples were prepared according to the constructed experimental design. Solutions of HWGA and SA were mixed at room temperature to the ratio specified for each run. The specified amount of acid was then added to the mixture, followed by the addition of the bFGF solution. The samples were placed on a rocker (BF-360, Biofree, Seoul, Korea) at 4 °C. They were then characterized by the method described in Section 2.3.1. for turbidity, encapsulation efficiency, Z-average, and PDI. The effect of the composition on each dependent response was evaluated based on Scheffe-type polynomial models as expressed by the equations below (Equations (2)–(5)) [16].
(2)$$\mathrm{Linear}:\;Y=\sum_{i=1}^{q}\beta_{i}x_{i}$$
(3)$$\mathrm{Quadratic}:\;Y=\sum_{i=1}^{q}\beta_{i}x_{i}+\sum \sum_{i=1}^{q}\beta_{ij}x_{i}x_{j}$$
(4)$$\mathrm{Special}\;\mathrm{Cubic}:\;Y=\sum_{i=1}^{q}\beta_{i}x_{i}+\sum \sum_{i<j}^{q}\beta_{ij}x_{i}x_{j}+\sum \sum_{i<j<k}^{q}\sum \beta_{ijk}x_{i}x_{j}x_{k}$$
(5)$$\mathrm{Full}\;\mathrm{Cubic}:\;Y=\sum_{i=1}^{q}\beta_{i}x_{i}\sum \sum_{i<j}^{q}\beta_{ij}x_{i}x_{j}+\sum \sum_{i<j}^{q}\delta_{ij}x_{i}x_{j}\left(x_{i}-x_{j}\right)+\sum \sum_{i<j<k}^{q}\beta_{ijk}x_{i}x_{j}x_{k}$$

### 2.5. In Vitro Cell Activity Study

#### 2.5.1. In Vitro Cell Viability Assay

For this experiment, freeze-dried bFGF coacervate samples were made with a 1:1 ratio of HWGA/SA (4 mg) and 10 ng of bFGF with 0.00025% drug loading (10 ng/4 mg). Human dermal fibroblasts (HDFs) were cultured in DMEM/low glucose supplemented with 10% FBS, penicillin (100 U/mL), and streptomycin (100 μg/mL) in a 5% CO_2_ incubator at 37 °C. For hyperglycemic conditions, glucose (25 mM) was added to the normal conditioning media and filtered using a syringe filter [11]. Cells were cultured at Passage 8 in a 100 mm dish until 80% confluence, then 1 × 10^4^ cells were seeded per well in a 24-well plate with normal (5 mM glucose) or hyperglycemic (25 mM glucose) conditioning media and were incubated at 37 °C for 24 h. After that, each well was washed once with DPBS and replenished with a media containing 0.5% FBS in each medium except for the positive group (10% FBS, 1 mL of normal or hyperglycemic media) and the bFGF group (10 ng/mL bFGF in 1 mL of 0.5% FBS normal or hyperglycemic media). Freeze-dried samples were put in the Transwell inserts, and 100 μL of the medium containing 0.5% FBS was added to rehydrate the samples. After these inserts were assembled into each well, the cells were incubated at 37 °C and 5% CO_2_ for 5 days. The HDF viability was determined by the CCK-8 assay. All results were obtained from measurements at 450 nm and normalized to the results of the negative group.

#### 2.5.2. In Vitro Cell Scratch Wound Healing Assay

Freeze-dried bFGF coacervate samples were made with a 1:1 ratio of HWGA to SA (4 mg) and 10 ng of bFGF with 0.00025% drug loading (10 ng/4 mg). The mid-line on the back of each well of the 24-well plates was marked with a marker pen and the plate bottom was coated with 0.1% gelatin [11]. The plates were incubated in a 37 °C incubator for 2 h and washed with DPBS. HDFs were cultured under normal or hyperglycemic conditions at Passage 8 in a 100 mm dish until 80% confluence; 1 × 10^5^ cells per well were incubated at 37 °C and 5% CO_2_ overnight. After that, autoclaved 200 μL pipette tips were used to scratch the cells at the mid-line of the well, and the wells were washed twice with DPBS to remove cell debris. Cells were treated with 1 mL of media containing 0.5% FBS and 10 μg/mL of mitomycin C and incubated at 37 °C for 2 h. After that, images were taken with a light microscope (Leica, Wetzlar, Germany) as the 0 h images after scratching. Freeze-dried samples were put in the Transwell inserts and 100 μL of media containing 0.5% FBS in DMEM/low glucose was added to the inserts. The incubation media (900 μL) contained 0.5% FBS, except for the pPositive (10% FBS, 1 mL) and bFGF (10 ng/mL bFGF in 0.5% FBS 1 mL) groups. The experimental procedures for the hyperglycemic conditions were the same as above, except that the wells were filled with various media containing 25 mM glucose. After the cells were incubated at 37 °C for 24 h, wound closure width was evaluated using Image J (National Institutes of Health, Bethesda, MD, USA). The wound healing effects of various samples were calculated as the ratio of the wound closure of each sample to that of the negative group as shown in the following Equation (6):Fold of wound area = {(A_0_ − A_24_)/A_0_}sample/{(A_0_ − A_24_)/A_0_}Negative(6)
where A_0_ is the wound area at 0 h and A_24_ is area of the wound after 24 h.

#### 2.5.3. PICP ELISA Assay

The supernatant media of cultured HDF cells under hyperglycemic conditions were harvested to 1.5 mL centrifuge tubes. Harvested media were concentrated in Amicon^®^ ultra–15 centrifugal filters (10,000 Mw) by centrifuging at 4000 rpm. An ELISA assay was carried out for the concentrated media according to the Aviva Systems Biology ELISA protocol.

### 2.6. Statistical Analysis

Data are expressed as the mean ± standard deviation (SD). Statistical differences were performed using one-way analysis of variance (one-way ANOVA) followed by Dunnett’s test with GraphPad Prism 5 software (version 5.01, GraphPad Software Inc., La Jolla, CA, USA). For all tests, *** *p* < 0.001, ** *p* < 0.01, * *p* < 0.05.

## 3. Results

### 3.1. Characterization of Gelatin

Circular dichroism is one of the useful methods for assessing the secondary structure, folding, and binding properties of proteins. As the spectra of proteins are dependent on their molecular conformation, CD spectra can be used to estimate the structure of gelatin [18]. HWGA showed a spectrum similar to that of collagen, whereas LWGA showed a spectrum similar to that of denatured collagen (Figure 1a), as reported previously [19]. LWGA had negative bands at 220–230 nm. The β-structure selection (BeStSel) method was used to estimate the percentage of secondary structures [20]. HWGA had a distorted helical structure and LWGA had about 32.8% helical structures and 67.2% antiparallel structures.

To measure the pH-dependent charge profile, the zeta potential was determined as a function of pH (Figure 1b). The zeta potential of HWGA was higher than that of LWGA at all pH values except at pH 10. The isoelectric point—the pH at which the zeta potential value crossed zero in these experimental conditions—was 8.5 and 6.2 for HWGA and LWGA, respectively.

### 3.2. Characterization of HWGA-SA bFGF Complex Coacervates

#### 3.2.1. Characterization and Morphology of HWGA-SA bFGF Complex Coacervates

The abbreviated names of various samples tested in the present study and their compositions are listed in Table 1. The characteristics of a representative coacervate are shown in Table 2.

The complex coacervates showed a milky colloidal phase–phase separation (Figure 2a). Encapsulation of FITC-bFGF was confirmed visually by fluorescence of the coacervates droplets under the fluorescence microscope (Figure 2b) [17]. The cross-sectional images of the freeze-dried coacervates sample showed a fine fibrous structure when examined with a SEM (Figure 2c).

#### 3.2.2. Comparison of HWGA-SA and LWGA-SA Complex Coacervates

Various samples shown in Table 1 were characterized in terms of their encapsulation efficiency (EE), trypsin digestion, and in vitro release as described below.

Encapsulation efficiency (EE) of FITC labeled bFGF

There was no difference in EE between HFCA (93.7%) and LFCA (92.7%).

Trypsin digestion assay

The trypsin digestion assay was used to compare the proteolytic susceptibility of various samples to protease. Trypsin served as a representative protease to screen the various compositions of LWGA-SA and HWGA-SA coacervates for their protective effects against degradation by proteases in the chronic wound sites [9,10,11]. The bFGF group, bFGF in solution, was completely digested by trypsin at the 2 h time point (Figure 3a,b). There was no difference between HFCC and HFCA (Figure 3a). HWGA-SA and LWGA-SA are compared in Figure 3b. The bFGF in the physical mixture of HWGA-SA or LWGA-SA was less digested by trypsin compared with the corresponding physical mixtures. The HWGA-SA coacervate system made with citric acid could protect the encapsulated bFGF better than the LWGA-SA coacervate system (Figure 3b). The data indicate that the HWGA-SA coacervate system can protect bFGF more effectively than the LWGA-SA coacervate system.

In vitro release assay

The release profiles of FITC-bFGF from various samples are shown in Figure 3c. In the case of LWGA-SA systems, the release percentage at the 4 h time point was 70.11 ± 8.79% and 64.80 ± 4.32% for LFM and LFCA, respectively, while it was only 34.04 ± 1.42% for LFCC. HWGA-SA systems showed patterns similar to those of LWGA-SA systems but with a wider range of controlled release profiles. In particular, HFCC had a slow sustained release profile for 24 h. The release percentage was 71.87 ± 7.38%, 61.41 ± 2.42%, and 15.44 ± 4.61% for HFM, HFCA, and HFCC, respectively.

#### 3.2.3. Phase Diagram of HWGA-SA Mixtures

Three different phase diagrams were constructed for three HWGA/SA ratios (1:0.5, 1:1, and 0.5:1), as a function of the acetic acid concentration (Figure 4). The pHΦ is the onset pH for the turbidity increase. The pHm (pH at the maximum optical density or turbidity) differed depending on the composition: the more SA, the higher the pHm. The Z-average minimum was observed at 5 to 7.5 mM acid for all three compositions. PDI values near or below 0.4 were at 5 to 12.5 mM acid for all three compositions.

### 3.3. Design of Experiments (DOE)

#### 3.3.1. Optimization of the Formulation by Response Surface Design

Various process parameters play important roles in the initiation, continuation, and termination of the coacervation process [9,10]. These are the temperature, the ionic strength and pH of the reaction medium, the polymer mixing ratios, the molecular weight, the total concentration, and charge densities. In the present study, HWGA was selected over LWGA based on the trypsin digestion results (Figure 3b). To optimize the HWGA-SA coacervate system with the DOE approach, the polymer ratios and the medium’s pH were selected as two independent variables based on the phase diagrams in Figure 4. Table 3 and Table 4 list the levels of independent variables and the desired ranges of responses, respectively. The optimal range of turbidity was set between 0.6 and 1.5, which correspond to the turbidity at pHΦ and the lowest turbidity value at pHm, respectively (Figure 4a). The minimum zeta size was chosen as the smallest zeta size of the three GA/SA ratios at pHΦ. The maximum size was set to 700 nm. The minimum PDI value was chosen as the PDI value at pHΦ. The maximum PDI was set to 0.7 because a PDI higher than 0.7 indicates a highly polydisperse system [21]. The desired EE was set to between 80 and 100%.

Table 5 shows the list of 15 experimental runs and the observed responses for each condition. A summary of each pertinent response model and the regression results is listed in Table 6.

R^2^ is the percentage of the response variable’s variation. The adjusted R^2^ is a modification of R^2^ and is explained by its relationship with the number of predictive variables in the model. These demonstrate how well the model adapts to the predicted data from the equation. The higher the R^2^ and the adjusted R^2^, the more the model is suitable for the data [17]. The predicted residual sum of square (PRESS) is the sum of squares of the prediction error [22]. When the lack of fit value is smaller than an α of 0.05, a model including higher terms such as interactions or quadratic terms may result in a better fit. Thus, the most suitable model is selected on the basis of high R^2^ and adjusted R^2^ values, a low PRESS value, and a high lack of fit value relative to the other models [17]. A full quadratic equation was selected as the suitable model for each response by Minitab 16 (Table 6 and Table 7).

#### 3.3.2. Contour Plot of Overlapping Dependent Responses 

The effects of two independent variables, HWGA (X_1_) and AA (X_3_), on each response are presented in representative contour and response surface plots (Figure 5). The level of SA (X_2_) was fixed at 200 for these plots.

Figure 6 shows two representative overlapped contour plots to define the design space for all four responses. The white area in Figure 6a represents the optimized condition of X_1_ and X_3_ at a fixed X_2_ of 200, which meets the defined ranges for all four responses shown in Table 4. Likewise, the white area in Figure 6b represents the optimized condition of X_2_ and X_3_ at a fixed X_1_ of 200. 

To evaluate the accuracy of the derived overlapped contour plots, two additional experiments were carried out with two compositions, C_1_ and C_2_, in the white area of the overlapped contour plots [23]. The prediction errors were between 1.38% and 17.87% for C_1_, and between 1.80%, and 5.44% for C_2_ (Table 8). These results confirmed the suitability of the full quadratic model for the optimization of HWGA-SA coacervates. The C_2_ composition was selected as the optimized composition of coacervates for further study.

### 3.4. In Vitro Cell Activity Study

#### 3.4.1. In Vitro HDF Viability Assay

A CCK-8 assay was performed to test the effect of various bFGF formulations on the viability of HDF under hyperglycemic conditions. For HWGA-SA coacervates, an optimized composition from DOE composition C_2_ in Table 8 was used. The positive control demonstrated that HDFs were cultured under good media conditions. Under both normal and hyperglycemic conditions, all test groups enhanced the cell viability compared with the negative group (Figure 7a). Under normal conditions, the FCA group showed significantly enhanced viability compared with the bFGF group. Under hyperglycemic conditions, the FM, FCA, and FCC groups significantly enhanced their viability compared with the bFGF group (Figure 7b). Procollagen Type I carboxy-terminal propeptide (PICP), which is derived from Type I collagen, has been identified as an indicator of Type I collagen synthesis in skin recovery [24]. The bFGF, HVC, and FM groups did not show any significant effect on procollagen synthesis (Figure 7c). In contrast, both the FCA and FCC groups significantly enhanced the biosynthesis of procollagen compared with the bFGF or FM group.

#### 3.4.2. In Vitro HDF Scratch Wound Assay

Cell migration is one of the most useful indicators of wound repair. In cutaneous wound healing, skin cells migrate from the wound edges into the wound for the recovery of the skin. Analysis of cell migration by scratching a wound in vitro is a useful assay for quantifying the effect of cell migratory capacity in response to an experimental drug treatment [25]. The wound healing in the positive control group under hyperglycemic conditions significantly decreased compared with the normal conditions (Figure 8c,d). This indicated that the hyperglycemic condition decelerates the migration activity of HDF. Under both normal and hyperglycemic conditions, the FM group did not affect cell migration, in contrast to the viability assay results. On the other hand, both the FCA and FCC groups significantly accelerated migration under hyperglycemia compared with the FM and bFGF groups (Figure 8d).

## 4. Discussion

bFGF is one of the endogenous growth factors which play an essential role in skin wound healing, promoting the proliferation and migration of various cell types (fibroblasts, keratinocytes, and endothelial cells) in a wound site [26,27]. They can enhance the remodeling of the extracellular matrix, involving granulation tissue formation and re-epithelization in the wound healing process [28,29]. However, clinical application of growth factors including bFGF has had limited efficacy in chronic wounds, partly due to degradation by elevated levels of proteases.

We evaluated two types of gelatins in the present study for the encapsulation of bFGF. Our hypothesis was that HWGA would be more effective than LWGA in encapsulating bFGF, because of its higher molecular weight, helical content, and positive charge. The HWGA-SA coacervate system exhibited better protection of bFGF from trypsin digestion and controlled release compared with the LWGA-SA coacervate system, supporting our hypothesis. This could be due to the stronger electrostatic and H-bonding interactions between HWGA and SA compared with between LWGA and SA [10]. These results suggest that HWGA-SA coacervates might be a better delivery system for bFGF compared with LWGA-SA.

On the other hand, optimization of the HWGA-SA coacervate system would be challenging because of its narrow coacervation range with an early onset of precipitation [10]. Therefore, we applied a DOE to optimize the HWGA-SA coacervates, with the optimal ranges of each response obtained from the phase diagrams. The Box–Benkhen response surface model was suitable for exploring quadratic response surfaces and constructing second-order polynomial models for optimizing three independent variables [23]. Each response contour plot was calculated using the equation of a full quadratic model, which was a better fitting model compared with the other three models. The white area in two overlapped contour plots (Figure 6) represents the design space that meets all four responses. The results of additional experiments with two different compositions, C_1_ and C_2_, supported the validity of the DOE results, confirming that the proposed model is useful for selecting the optimal composition of HWGA-SA coacervates. In particular, C_2_ had a low percentage of prediction error (Table 8); therefore, it was selected as the optimized composition for in vitro studies.

Increased viability in the in vitro HDF model was related to the increased capacity of stimulating cell proliferation [30]. The much lower cell viability of the bFGF group under hyperglycemia than under normal conditions indicates a significant loss of bFGF activity in the solution under hyperglycemia, which is consistent with our previous findings with EGF [11]. All three compositions (HFM, HFCA, and HFCC) exhibited significantly enhanced cell viability activity relative to the bFGF group under hyperglycemic conditions, but not under normal conditions. This finding also demonstrates the importance of the delivery mode for bFGF under hyperglycemia. Complex coacervates HFCA and HFCC enhanced the activity of encapsulated bFGF in terms of not only HDF cell viability but also procollagen synthesis (Figure 7c). The scratch wound assay results were consistent with the PICP results (Figure 8). Under hyperglycemic conditions, the wound closure of the positive control group was significantly lower than that of the normal group, indicating that the hyperglycemic condition reduced the migration activity of fibroblasts (Figure 8c,d). These in vitro cell test results support our hypothesis that the activity of bFGF would be enhanced by encapsulating in HWGA-SA coacervates.

In summary, DOE was successfully applied to optimize the HWGA-SA coacervates’ DDS for the bFGF delivery system. Among various compositions, HFCA (freeze-dried HWGA-SA (1:1) coacervates encapsulating bFGF, prepared with acetic acid) is proposed as a potential therapeutic candidate for treating DFU. Future studies for clinical translation of the optimized HFCA would include in vivo studies in a diabetic wound model and storage stability tests. Cryoprotectant screening to improve the stability of freeze-dried coacervates would be another study before the final selection of a development candidate. Another interesting research subject would be to compare HFCA with the previously proposed EGF containing LWGA/SA coacervates, and an evaluation of coacervates encapsulating both EGF and FGF for potential synergy.

## 5. Conclusions

The HWGA-SA coacervate system was chosen over LWGA-SA as a DDS for bFGF because it was more effective in protecting bFGF from trypsin digestion. HGWA-SA systems with a narrow coacervation phase were successfully optimized by DOE using the polymer ratio and the medium’s pH as independent variables, and the turbidity, size, PDI, and encapsulation efficiency as the dependent responses. Full quadratic equations were used to construct overlapping dependent responses contour plots and design spaces. An optimized coacervate (HWGA:SA = 1:1, acidified with acetic acid, polymer: bFGF ratio = 4 mg/10 μg) effectively protected bFGF from trypsin digestion and showed controlled release. Moreover, bFGF complex coacervates enhanced the viability, collagen synthesis, and migration of HDF cells under hyperglycemic conditions. These results support the possibility that an optimized HWGA-SA bFGF complex coacervate system has potential as a novel therapeutic modality for diabetic foot ulcers. Further studies necessary for clinical translation of the present results would be in vivo studies in a diabetic wound model and storage stability tests.

## Figures and Tables

**Figure 1 pharmaceutics-13-02112-f001:**
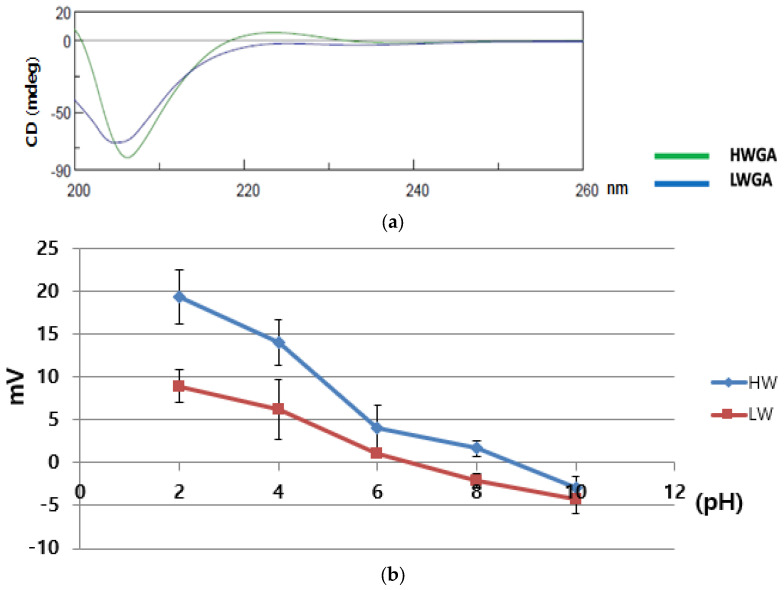
Characterization of Gelatin A. (**a**) Circular dichroism spectra of HWGA and LWGA solutions (5 × 10^−2^ g/L). (**b**) The pH-dependent zeta potential of 1% HWGA and 1% LWGA.

**Figure 2 pharmaceutics-13-02112-f002:**
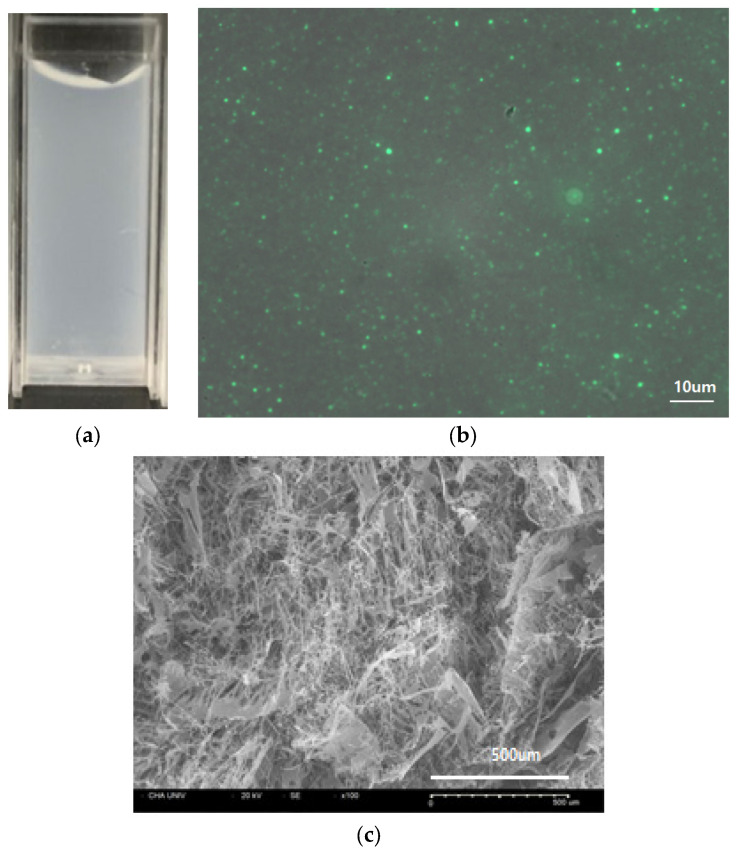
Morphology of HWGA-SA bFGF complex coacervates. (**a**) Photographic image of bFGF coacervates, (**b**) fluorescence microscope image of FITC-bFGF coacervate, and (**c**) SEM image of a cross-section of freeze-dried coacervate.

**Figure 3 pharmaceutics-13-02112-f003:**
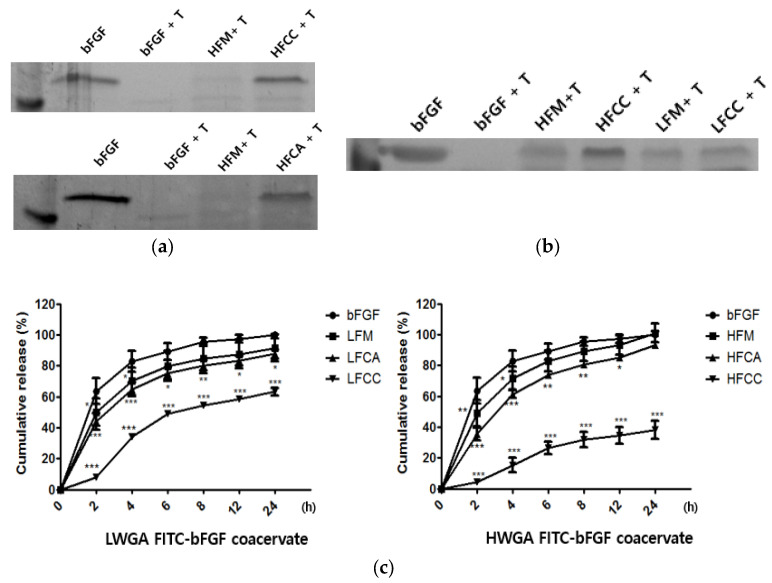
Comparison of LWGA-SA and HWGA-SA coacervates with corresponding free solutions and physical mixtures. (**a**,**b**) Trypsin digestion assay of bFGF in various samples after incubation with trypsin for 2 h. (**c**) Quantitative release profiles of FITC-bFGF from various systems. *** *p* < 0.001, ** *p* < 0.01, * *p* < 0.05.

**Figure 4 pharmaceutics-13-02112-f004:**
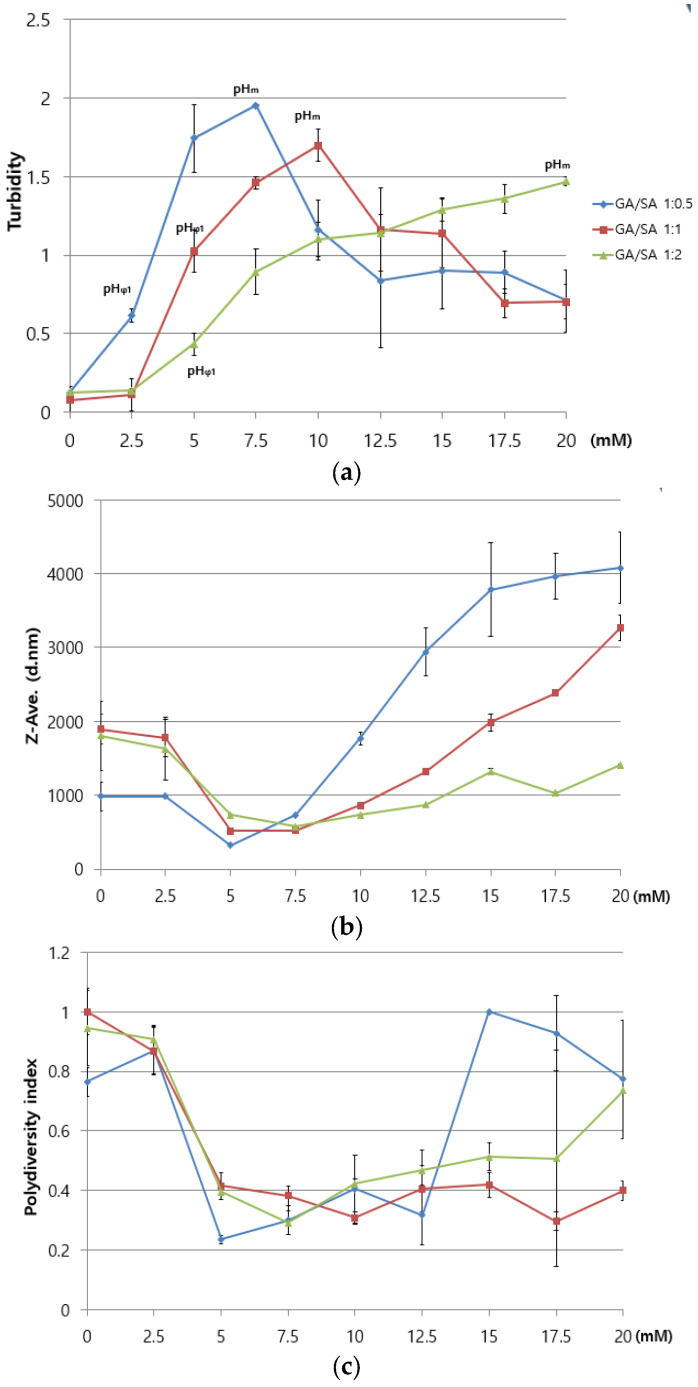
Phase diagrams of HWGA-SA mixtures: (**a**) turbidity, (**b**) Z-average, and (**c**) polydispersity index (PDI) as a function of the acetic acid concentration (mM). pHΦ: onset pH for phase separation; pHm: pH of maximum optical density.

**Figure 5 pharmaceutics-13-02112-f005:**
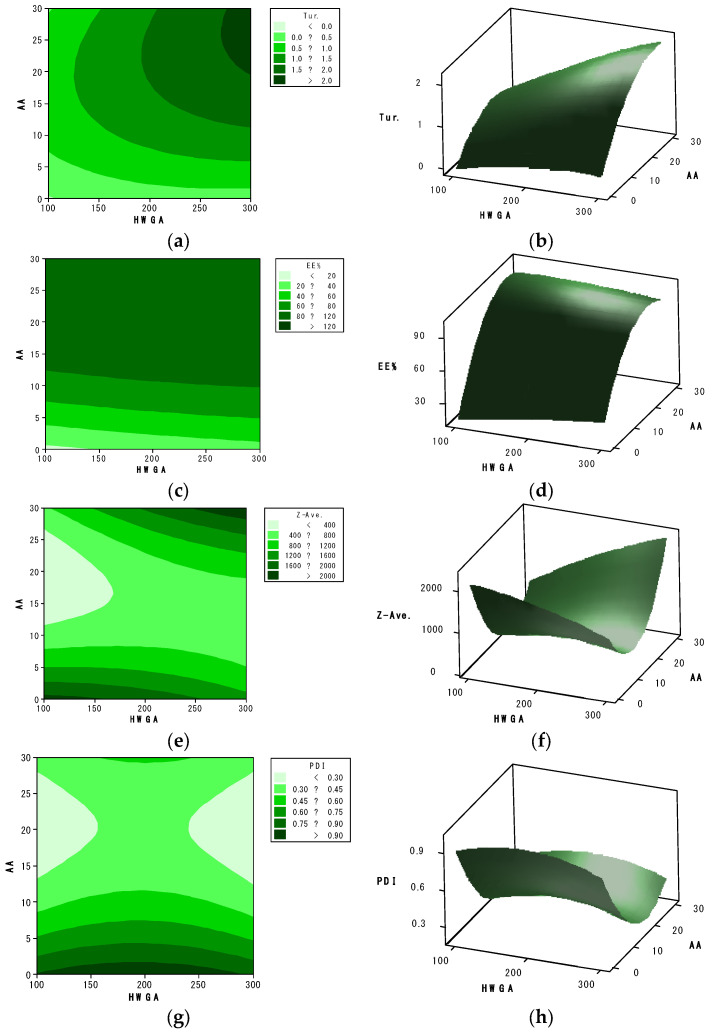
Contour plots (**a**,**c**,**e**,**g**) and response surface plots (**b**,**d**,**f**,**h**) of turbidity (**a**,**b**), encapsulation efficiency (**c**,**d**), particle size (**e**,**f**), and polydispersity index (**g**,**h**). HWGA and AA represent the volumes (μL) of the 1% HWGA (X_1_) and 0.5 mM acetic acid solutions (X_3_), respectively. SA (X_2_) was fixed at 200 μL.

**Figure 6 pharmaceutics-13-02112-f006:**
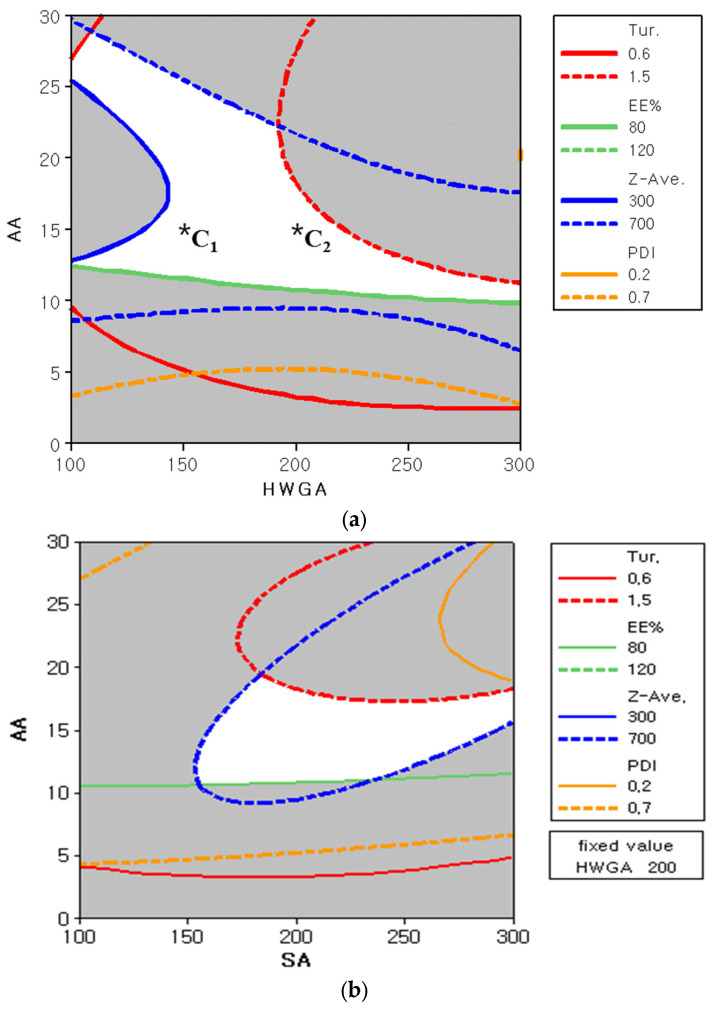
Overlapped contour plots of turbidity, encapsulation efficiency, particle size, and polydispersity index to define the design space of two independent variables with (**a**) a fixed SA value at 200 and (**b**) a fixed HWGA value at 200. *C_1_: GA/SA/AA = 150/200/15. *C_2_: GA/SA/AA = 200/200/15.

**Figure 7 pharmaceutics-13-02112-f007:**
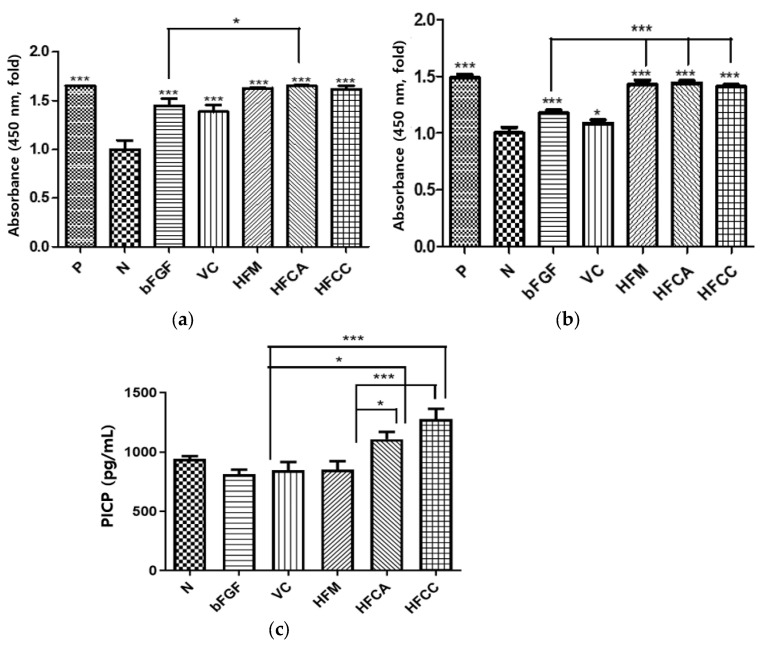
In vitro activity assay of HWGA-SA bFGF coacervates at 5 days (*n* = 3). (**a**) HDF viability assay under normal conditions and (**b**) hyperglycemic conditions, and (**c**) PICP ELISA assay under hyperglycemic conditions. *** *p <* 0.001, * *p <* 0.05.

**Figure 8 pharmaceutics-13-02112-f008:**
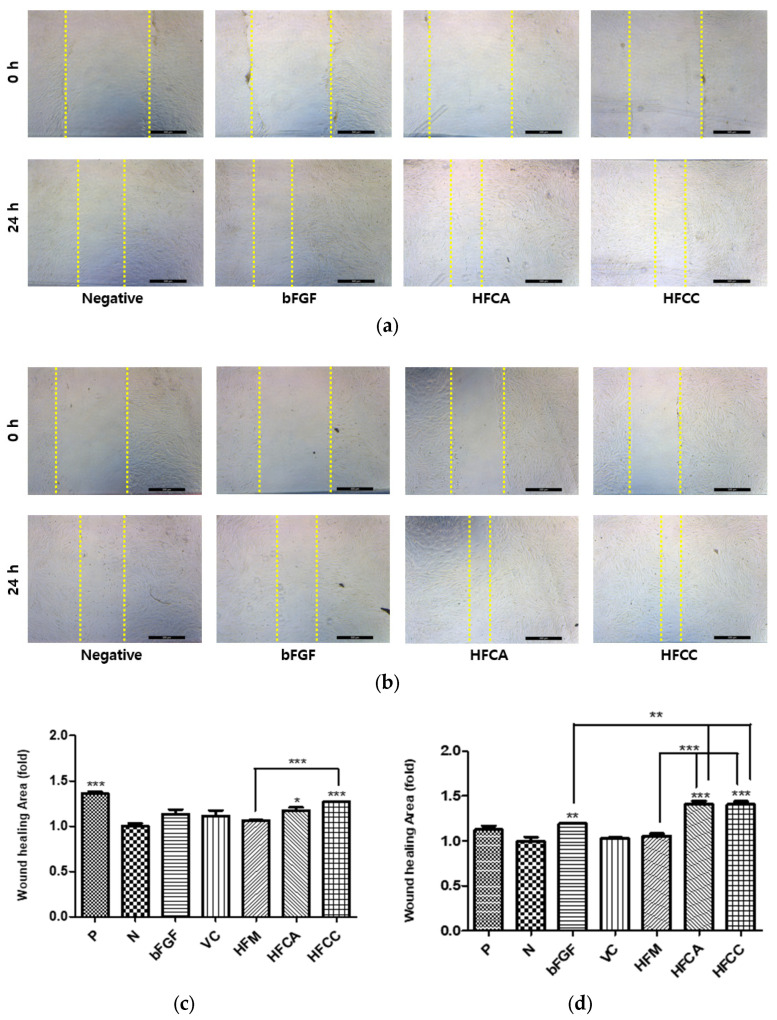
Representative bright-field images of the HDF scratch wound assay (**a**) under normal conditions and (**b**) hyperglycemic conditions. Quantitative statistical analysis (**c**) under normal conditions and (**d**) hyperglycemic conditions after 24 h (*n* = 3). Scale bars = 500 µm. *** *p* < 0.001, ** *p* < 0.01, * *p* < 0.05.

**Table 1 pharmaceutics-13-02112-t001:** Sample name abbreviations and their compositions.

Abbreviation	Composition
Negative Group	No treatment
bFGF Group	bFGF in solution
VC	Vehicle control
VCA	Freeze-dried vehicle control group containing acetic acid (empty HWGA-SA coacervates prepared with acetic acid).
VCC	Freeze-dried vehicle control group containing citric acid (empty HWGA-SA coacervates prepared with citric acid).
HFM	Freeze-dried physical mixture of HWGA, SA, and bFGF
HFCA	Freeze-dried HWGA-SA coacervates encapsulating bFGF, prepared with acetic acid
HFCC	Freeze-dried HWGA-SA coacervates encapsulating bFGF, prepared with citric acid
LFM	Freeze-dried physical mixture of LWGA, SA, and bFGF
LFCA	Freeze-dried LWGA-SA coacervates encapsulating bFGF, prepared with acetic acid
LFCC	Freeze-dried LWGA-SA coacervates encapsulating bFGF, prepared with citric acid

**Table 2 pharmaceutics-13-02112-t002:** Characteristics of a representative complex coacervate encapsulating bFGF.

HWGA/SA Ratio	Acid Vol. (μL) *	pH	Turbidity	EE%	Z-Ave. (nm)	PDI	Zeta- Potential (mV)
1:1	15	4.34 (±0.06)	1.462 (±0.03)	93.68 (±0.2)	526.4 (±6.87)	0.38 (±0.02)	−47.3 (±0.5)

* Volume of the acetic acid solution (25 mM) added to 1 mL of a mixed solution of HWGA (1%) and SA (1%).

**Table 3 pharmaceutics-13-02112-t003:** Independent variables and their levels in the response surface experimental design.

Independent Variables	Level	
	Low	High
X_1_ (μL): 1% HWGA	100	300
X_2_ (μL): 1% SA	100	300
X_3_ (μL): 0.5 mM acetic acid	0	30

**Table 4 pharmaceutics-13-02112-t004:** Desired ranges of dependent responses.

Dependent Responses	Goal	Lower Limit	Upper Limit
Y_1_: Tur.	Fit in range	0.6	1.5
Y_2_: EE (%)	Fit in range	80	100
Y_3_: Z-Ave. (d.nm)	Fit in range	300	700
Y_4_: PDI	Fit in range	0.2	0.7

**Table 5 pharmaceutics-13-02112-t005:** The Box–Behnken experimental designs and responses.

Run	Independent Variables			Dependent Responses *			
	X_1_	X_2_	X_3_	Y_1_	Y_2_	Y_3_	Y_4_
12	200	300	30	1.6492	89.5774	925.50	0.20533
2	300	100	15	1.7953	94.7828	1867.67	0.28300
13	200	200	15	1.3577	93.8718	504.37	0.39400
14	200	200	15	1.3192	92.0561	470.70	0.35100
9	200	100	0	0.1229	26.9478	1060.73	0.86367
5	100	200	0	0.1301	11.5720	2785.67	1.00000
8	300	200	30	1.9769	91.5266	1673.00	0.26500
6	300	200	0	0.1261	33.2412	1576.00	0.98600
11	200	100	30	0.9860	92.0323	4384.67	1.00000
10	200	300	0	0.1357	23.8413	2876.00	1.00000
15	200	200	15	1.5181	90.4045	498.97	0.35900
3	100	300	15	0.4641	88.3662	678.50	0.28600
7	100	200	30	0.6018	92.9564	503.47	0.23867
4	300	300	15	1.8250	87.2374	832.00	0.24267
1	100	100	15	0.5082	89.7231	360.17	0.31467

* Y_1_: turbidity; Y_2_: encapsulation efficiency; Y_3_: Z-average; Y_4_: polydispersity index.

**Table 6 pharmaceutics-13-02112-t006:** Regression results of the response surface models’ fit.

**Responses**	**Model**	**R^2^**	**Adjusted R^2^**	**SD**	**Lack of Fit**	**Press**
Turbidity	Full quadratic	95.85	88.39	0.239	0.114	4.270
EE		99.63	98.96	3.176	0.173	724.411
Z-Ave.		89.13	69.55	640.732	0.000	32,833,974
PDI		91.70	76.76	0.161	0.012	2.070
Responses	Model	R^2^	Adjusted R^2^	SD	Lack of fit	Press
Turbidity	Quadratic	87.40	50.18	0.330	0.075	3.435
EE		98.57	94.37	4.918	0.090	763.539
Z-Ave.		42.36	0.00	1166.25	0.000	43,523,153
PDI		77.87	11.60	0.208	0.009	1.388
Responses	Model	R^2^	Adjusted R^2^	SD	Lack of fit	Press
Turbidity	Special Cubic	78.55	20.79	0.430	0.044	5.462
EE		69.21	0.00	22.848	0.004	17,737.5
Z-Ave.		52.64	0.00	1057.18	0.000	37,269,026
PDI		54.56	0.00	0.299	0.004	3.508
Responses	Model	R^2^	Adjusted R^2^	SD	Lack of fit	Press
Turbidity	Linear	70.10	61.94	0.433	0.048	N/A
EE		68.16	59.47	1270.990	0.006	N/A
Z-Ave.		5.87	0.00	640.732	0.000	N/A
PDI		40.73	24.56	0.291	0.00	N/A

**Table 7 pharmaceutics-13-02112-t007:** The equations of the full quadratic model for each dependent response.

Responses	Equation
Turbidity	Y_1_ = −0.828567 + 0.00650529X_1_ + 0.00354117X_2_ + 0.0457964X_3_ + 1.845 × 10^−6^X_1_X_2_ + 0.000229850X_1_X_3_ + 0.000108400X_2_X_3_ − 1.32454 × 10^−5^X_1_^2^ − 1.7729 × 10^−5^X_2_^2^ − 0.00247624X_3_^2^
EE	Y_2 =_ −3.23038 + 0.1760285X_1_ + 0.0373900X_2_ + 7.44996X_3_ − 1.54714 × 10^−4^X_1_X_2_ − 0.00384982X_1_X_3_ + 0.000108598X_2_X_3_ − 1.42955 × 10^−4^X_1_^2^ − 6.53892 × 10^−5^X_2_^2^ − 0.148254X_3_^2^
Z-Ave.	Y_3_ = 1009.41 + 7.52714X_1_ − 5.40407X_2_ − 78.2790X_3_ − 0.0338500X_1_X_2_ + 0.396533X_1_X_3_ − 0.879072X_2_X_3_ − 0.0116976X_1_^2^ + 0.0560215X_2_^2^ + 5.60073X_3_^2^
PDI	Y_4_ = 0.400458 + 0.00450417X_1_ + 0.000308333X_2_ − 0.0374889X_3_ − 2.9667 × 10^−7^X_1_X_2_ − 6.72222 × 10^−6^X_1_X_3_ − 1.55167 × 10^−4^X_2_X_3_ − 1.15625 × 10^−5^X_1_^2^ − 2.92083 × 10^−6^X_2_^2^ + 0.00164463X_3_^2^

**Table 8 pharmaceutics-13-02112-t008:** The additional test of prediction errors.

Responses	Model Prediction Value (Y) (X_1_, X_2_, X_3_)		Experimental Value (Y’) (X_1_, X_2_, X_3_)		Percentage of Prediction Error (|Y−Y’|/Y%)	
	C_1_	C_2_	C_1_	C_2_	C_1_	C_2_
Turbidity (Y_1_)	1.11614	1.39816	1.101	1.4238	1.38	1.80
EE (Y_2_)	90.3272	92.2060	95.89	95.03	6.16	3.06
Z-Ave. (Y_3_)	361.276	490.948	439.9	519.2	17.87	5.44
PDI (Y_4_)	0.343038	0.367753	0.298	0.353	15.11	4.18

C_1_: composition 1—GA/SA/AA: 150/200/15. C_2_: composition 2—GA/SA/AA: 200/200/15.

## Data Availability

Data is contained within the article.

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
