# Peer review of "Gelatin-Alginate Coacervates Optimized by DOE to Improve Delivery of bFGF for Wound Healing"

_pharmaceutics, 2021, doi:10.3390/pharmaceutics13122112_

Round 1

Reviewer 1 Report

This is a well-organized and well-illustrated paper, has an important clinical message, and should be of great interest to the readers. This paper reported the development of gelatin-alginate coacervates optimized by the design of experiment to enhance the deliery of fibroblast growth factor for wound healing. This research opens up new possibilities in treating diabetic foot ulcers. This manuscript deserves publication after addressing the minor issues cited below.

  1. I suggest the authors to cite some important papers that focussed on delivery of fibroblast growth factors using various formulations in the introduction such as:  Small. 2008 Oct; 4(10): 1769–1777, 
    Tissue Eng Regen Med. 2016 May;10(5):418-27
  2. I suggest the authors to include the curent challenges or limitations associated with the clinical translation of the proposed materials in lines 503-504.
  3. In the conclusion section, i sugest the authors to summarize all the resullts presented in the paper.

Reviewer 2 Report

I read this paper carefully; the scientific idea could be interesting, but the study and the drafting of the paper are not well structured. The description of the methods and scientific language are not clear (sometimes inadequate) and do not help the reader to understand the text and setting of the work. There are also inaccuracies in punctuation (example line 15). Sometimes unnecessary details are described (e.g. the pipette colour, line 200)

The introduction is too long and should be shortened.

The dissolution test is not described correctly and it is not clear how it was done. I don't know the "transwell inserts" (line 158) but I don't think they are tools for performing dissolution tests. The authors should be more precise and report official methods (e.g. official pharmacopoeias) or bibliographic references.

2.5.1 The in vitro cell viability test is normally performed by the MTT test. The test described in this paper is carried out in hyperglycemic conditions. Authors should explain if this test is scientifically accepted and used or whether it was developed by themselves.

Line 182, line 199: the authors should explain the unit of measure π, line 335 or pHΦ

2.5.2. Even the scratch test is performed in conditions of hyperglycemia (?); is this a universal accepted method or it has been developed by the authors?

3.1. why is gelatine characterized?

3.2.2. the structuring of this part of the work is not understandable.

Line 284: tripsin digestion assay? Why?

Line 304: “in vitro release assay” - it is not clear how it was carried out.

Line 439: “discussion” -  the discussion is written in a very confusing and inappropriate way. There are introduction parts (e.g. lines 440-447) and parts that describe the experiments.

The conclusions are very short and not supported by experimental data.

Reviewer 3 Report

  1. Fabrication of cocervates: What is the criteria of selection of concentration of 1% LWGA (w/w) or 1% HWGA (w/w) and 1% SA (w/w).
  2. Give the reference for the method of preparation.
  3. Freeze drying: Is cryprotectant used for freeze drying. If used mention the concentration.
  4. Give the equation in equation format with equation number.
  5. Table: Third factor X3 with thwe upper and lower level is missing.
  6. Run 12 and Run 2 having total concentration of 500 and 400. But at low volume the size is very big. 
  7. In what unit the variables X1, X2, X3 taken
  8. Explain in detail about Optimization of formulation by response surface design. THe present explanation is not sufficient.
  9. The manuscript required spelling and grammer check. 

Round 2

Reviewer 3 Report

Accept